# Defoliation-induced changes in foliage quality may trigger broad-scale insect outbreaks

Louis De Grandpré [1✉], Maryse Marchand [1], Daniel D. Kneeshaw [2], David Paré[1], Dominique Boucher[1], Stéphane Bourassa[1], David Gervais[1], Martin Simard [3], Jacob M. Griffin [4] & Deepa S. Pureswaran[5]

Top-down effects, like predation, are drivers of insect outbreaks, but bottom-up effects, like host nutritional quality, also influence outbreaks and could in turn be altered by insect-caused defoliation. We evaluated the prediction that herbivory leads to a positive feedback on outbreak severity as nutrient concentration in plant tissues increases through improved soil nutrient availability from frass and litter deposition. Over seven years of a spruce budworm outbreak, we quantified litter nutrient fluxes, soil nitrogen availability, and host tree foliar nutrient status along a forest susceptibility gradient. As the outbreak progressed, both soil nutrient fluxes and availability increased which, in turn, improved foliage quality in surviving host trees. This is consistent with boosted insect fitness and increased population density and defoliation as outbreaks grow. Our results suggest that a positive bottom-up feedback to forest ecosystems from defoliation may result in conditions favorable to self-amplifying population dynamics in insect herbivores that can contribute to driving broad-scale outbreaks.

[1] Natural Resources Canada, Canadian Forest Service, Laurentian Forestry Centre, Quebec City, QC, Canada. [2] Centre for Forest Research, Department of Biology, Université du Québec à Montréal, Montréal, QC, Canada. [3] Department of Geography, Centre for Forest Research, Laval University, Quebec City, QC, Canada. [4] Department of Biological Sciences, Edgewood College, Madison, WI, USA. [5] Natural Resources Canada, Canadian Forest Service, Atlantic Forestry Centre, Fredericton, NB, Canada. ✉email: louis.degrandpre@canada.ca

nsects are a structuring force in forest ecosystems, influencing the properties of plant and animal communities, soil processes, and their interactions[1–4]. Defoliators increase nutrient transfer from plants to soil via frass, insect corpses, and damaged green foliage that drops to the forest floor[5–8]. They enhance nutrient mineralization by changing soil micro-environmental conditions generally favoring microbial activity, as well as the quality of litter for decomposers[9–12]. It has in fact been suggested that insect herbivores play an underestimated role in contributing to the nutrient cycling of unproductive boreal forest ecosystems[3].

Nutrient cycling in the boreal forest is slow, despite abundant quantities of nitrogen (N) and phosphorus (P) in forest soils. These elements are mostly bound in organic molecules whose decomposition is slow, thereby limiting plant uptake and greatly restricting forest growth[13,14]. Consequently, the addition of more readily accessible nutrients generally results in enhanced productivity in boreal forest stands[15].

Under natural conditions, disturbances such as fire and insect outbreaks release nutrients that are immobilized in plant tissue and create conditions that favor increased mineralization of soil nutrients[16–18]. Increase in soil temperature following disturbance promotes microbial activity and results in a positive effect on nutrient cycling[19–21]. Such major shifts in nutrient fluxes are usually observed following broad-scale host-tree mortality during outbreaks[18,22,23]. In defoliation episodes, it is likely that surviving host trees benefit from the greater supply of available soil nutrients as the canopy progressively opens and forest environmental conditions, such as soil temperature, are modified[5,10,12,20,24]. In fact, ecosystems affected by insect defoliation lose very little nitrogen, as both surviving and non-affected plants as well as soil microbes rapidly immobilize the newly released nutrients[5,6,25].

In addition to influencing ecosystem nutrient dynamics, it has been proposed that herbivores could also respond to these changes by increasing their fitness and subsequently population growth[2]. However, despite early work suggesting that bottom-up forces were a factor controlling spruce budworm (SBW) outbreaks, this idea was discredited in the SBW literature based on population models because outbreaks collapsed even when large patches of host species were still available in the forest[26]. Only recently has it been shown that specialist herbivore fitness may depend on both top-down (e.g., natural enemies) and bottom-up forces[27]. For insect defoliators, nitrogen is the main factor limiting growth, thus, host nutritional quality could be the feedback mechanism that allows populations to grow quickly as soon as top-down pressure is relaxed[26]. It was hypothesized that coniferous forests are nutritional deserts that cannot support SBW population growth until some conditions, like a drought stress, raised the nutritional status of host trees[28,29]. Population growth of SBW was positively correlated across a broad-scale, with years of highly nutritious cone production in host trees[30]. In addition, we propose that a positive feedback loop between the herbivore, soil, and plants[2,6] may be key to local population build-up. Changing environmental conditions with the progression of a defoliator outbreak, including increased light in the understory and soil nutrient availability over several years, could gradually alter tree nutritional status[5,31]. Such processes would be of particular importance in forest ecosystems where nutrient availability is highly limited.

In 2006, a few hot spots of defoliation by the SBW were observed in northeastern Quebec[32,33]. In the years that followed, the defoliated area increased in size and severity and populations of the SBW reached outbreak levels. In this early phase of the outbreak, we installed permanent plots in stands of various host canopy composition (from stands dominated by balsam fir (*Abies balsamea* (L.) Mill.), the main host, to those dominated by back spruce (*Picea mariana* (Mill.) B.S.P.), a secondary host) to

monitor the effects of the outbreak on the forest ecosystem. Given the unique opportunity to track changes in the ecosystem from the onset of the outbreak to the point where mortality began, the objective of this study was to evaluate the effect of multi-year SBW defoliation on nutrient dynamics. We hypothesized that as SBW defoliation progresses: (1) nutrient fluxes and concentration in litterfall would increase, (2) the increase in litter nutrient fluxes and concentration would be more pronounced in forest stands dominated by the primary host of the SBW, balsam fir, and (3) that in combination with increased frass and needle inputs, the rise in soil temperature would trigger changes in soil nutrient dynamics, which would increase host foliar nutrient concentration, an effect that would be stronger for the primary host balsam fir than black spruce. Using linear mixed-effects models, we confirmed our three main hypotheses. Soil nutrient fluxes and availability increased with outbreak progression which, in turn, improved foliage quality in surviving host trees. These effects could create a positive feedback loop on further insect population growth and be a factor that triggers and sustains outbreaks.

## Results

**Overview of the outbreak**. Although annual defoliation patterns were similar and their peaks synchronous among stand composition types, balsam fir-dominated stands sustained significantly more defoliation than black spruce-dominated stands (Fig. 1a). SBW population density followed a pattern similar to annual defoliation, with a notable peak in 2014 and a sharp drop in 2015 (Fig. 1b). However, significantly more fourth-instar larvae (L4) were found in fir-dominated and mixed stands compared to black spruce-dominated stands (Fig. 1b). Mean annual litterfall (combination of host conifer needles, deciduous leaves, and SBW frass) followed the same temporal pattern across stand types, but there was significantly more litterfall in balsam fir-dominated stands compared to black spruce-dominated stands (Fig. 1c). A decrease in litterfall occurred in 2015 and 2016 though markedly less than what was observed for defoliation and larval abundance. Cumulative tree mortality abruptly increased following the 2014 defoliation peak for both balsam fir-dominated and mixed stand types and was significantly higher than in black spruce-dominated stands (Fig. 1d). Details on these regression model results can be found in Supplementary Table 1.

**Nutrient fluxes via litterfall**. Elemental fluxes from litterfall increased for carbon and for all nutrients. Carbon (C) fluxes more than doubled when comparing 2011 to the defoliation peak of 2017 (Table 1). Nitrogen (N) and phosphorus (P) fluxes increased from four to more than five-fold (Table 1). P fluxes followed similar trends among stand types but at levels 10 times lower than for N. Potassium (K) fluxes increased between two to four-fold (Table 1).

Nitrogen concentration (g kg$^{-1}$) significantly increased as the outbreak progressed in all stands and litter types, with the exception of deciduous leaf litter in spruce-dominated stands, which did not increase significantly (Fig. 2a). The rate of N increase by litter compartments was significantly higher when balsam fir was dominant in the stand than in other stand types (Fig. 2a). Phosphorus concentration significantly increased in all litter types as the outbreak progressed, and the rate of increase in P concentration was significantly higher in fir-dominated and mixed stands than in spruce-dominated stands (Fig. 2b). Potassium concentration generally decreased with time in needle and leaf litters (Fig. 2c). As the outbreak progressed, the decrease in litter K concentration was greater in spruce-dominated stands compared to other stand compositions (Fig. 2c).

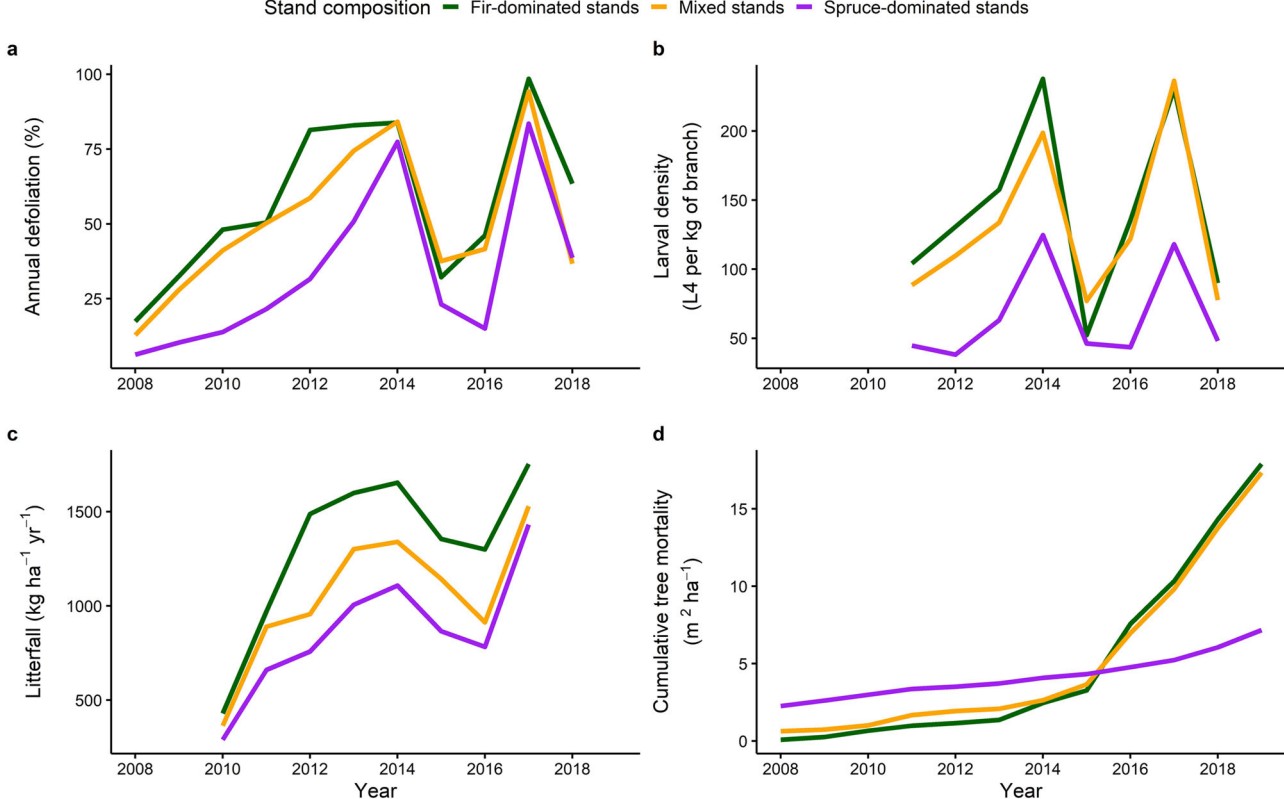

**Fig. 1 Overview of SBW outbreak progression among three forest composition types.** Annual defoliation (**a**), larval density (**b**), annual litterfall (**c**), and cumulative tree mortality (**d**) with spruce budworm outbreak progression among three forest composition types. Annual defoliation and larval density were averaged from the sampled branches of host trees at the stand level. Litterfall represents the total litterfall (needle and leaf litter, SBW frass) in the 21 traps in each stand, while mortality represents the cumulative annual dead basal area of SBW hosts within each stand. Stand level values were then averaged by forest composition types.

Nitrogen and P fluxes (kg ha$^{-1}$) in needle litter increased significantly with time in all stand composition types but the rate of increase was higher in fir-dominated stands compared to either mixed and spruce-dominated stands (Fig. 3a, b). Frass N and P fluxes also significantly increased as the outbreak progressed in all stand types. Frass litter K flux increased significantly with time (Fig. 3c). More details on the effects of outbreak progression on nutrient concentration and fluxes can be found in Supplementary Tables 2 and 3. There were some differences in nutrient concentration and fluxes for each litter type among stand types at the beginning of the study (Supplementary Tables 4 and 5).

**Nutrient concentration and C:N ratio in live host tree needles.**
Live needle N concentrations increased significantly faster, as the outbreak progressed, in balsam fir than in black spruce foliage (Fig. 4a). Most slopes were significantly different from zero and showed an increase in N concentration as the outbreak progressed (Fig. 4a). The rate of increase in P concentration with time was higher in needles of balsam fir than of black spruce and did not differ among stand composition types (Fig. 4b). As the outbreak progressed, K concentration in SBW host needles showed a significant decrease, which was similar among stand types but significantly sharper in black spruce compared to balsam fir (Fig. 4c). Needle C:N ratio decreased significantly with time for both tree species in all stand composition types although the decline was greater for fir needles (Fig. 4d). See Supplementary Table 6 for more information on progression of foliar concentrations with time. Differences in initial needle nutrient concentration are shown in Supplementary Table 7.

**Soil temperature and inorganhic nitrogen.** Soil temperature increased significantly with increasing defoliation, month and stand type. June and July warming rates did not significantly differ among stand types as defoliation progressed (Supplementary Table 8 and Fig. 5). In August, the warming rate with defoliation was significantly lower in spruce-dominated stands than other stand types (Fig. 5).

Soil inorganic nitrogen (SIN) also significantly increased with defoliation (0.02, $t = 4.78$, $p < 0.001$), as well as with increasing local abundance of balsam fir (0.01, $t = 2.63$, $p = 0.008$; Fig. 6). SIN decreased with increasing basal area of live trees ($-0.03$, $t = -3.75$, $p < 0.001$; Fig. 6).

## Discussion

This study provides a unique long-term perspective on the impact of an insect outbreak on forest nutrient dynamics and insights on how defoliating insects may positively influence fitness and population growth of future generations by increasing food resource quality[2,34]. All three of our hypotheses were verified: as SBW populations built up over the seven years of our study (2011–2017), annual defoliation, litterfall, and associated nutrient fluxes increased. Defoliation as well as changes in nutrient fluxes in litterfall were greater in stands with a greater proportion of fir. Finally, foliage nutrient concentration and soil inorganic nitrogen availability also increased over the course of the outbreak. The only exception is K for which the concentration in live needles and deciduous leaf litter decreased over the course of the outbreak. Potassium is an element that is cycled wastefully at the plant level, it is easily leached from foliage and much greater

**Table 1 Extent of yearly values for litterfall (needles, leaves, and spruce budworm frass) carbon and nutrients fluxes (kg ha⁻¹ yr⁻¹) during a spruce budworm outbreak in northeastern Canada.**

| Stand type | Carbon | | | Nitrogen | | | Phosphorus | | | Potassium | | |
|---|---|---|---|---|---|---|---|---|---|---|---|---|
| | Min | Max | Ratio | Min | Max | Ratio | Min | Max | Ratio | Min | Max | Ratio |
| Bf-stand[a] | 359 | 1274 | 3.6 | 6.2 | 34.0 | 5.5 | 0.7 | 3.4 | 4.9 | 0.8 | 2.2 | 2.8 |
| Bs-stand[b] | 337 | 890 | 2.6 | 3.2 | 12.9 | 4.0 | 0.3 | 1.6 | 5.3 | 0.4 | 0.9 | 2.3 |
| Mixed[c] | 410 | 960 | 2.3 | 5.1 | 22.9 | 4.5 | 0.5 | 2.1 | 4.2 | 0.5 | 1.5 | 3.0 |

Bs: Black spruce-dominated stands, where basal area (m² ha⁻¹) of black spruce represents > 75% of total coniferous tree basal area in the stand‡.Mixed stands, where balsam fir and black spruce each represent between 25% and 75% of total coniferous basal area.
[a]Minimum values occurred in 2011 or 2012 and maximum values at defoliation peaks (2014 or 2017).
[b]Bf: Balsam fir-dominated stands, where basal area (m² ha⁻¹) of balsam fir represents > 75% of total coniferous tree basal area in the stand.
[c]Bs: Black spruce-dominated stands, where basal area (m² ha⁻¹) of black spruce represents > 75% of total coniferous tree basal area in the stand.

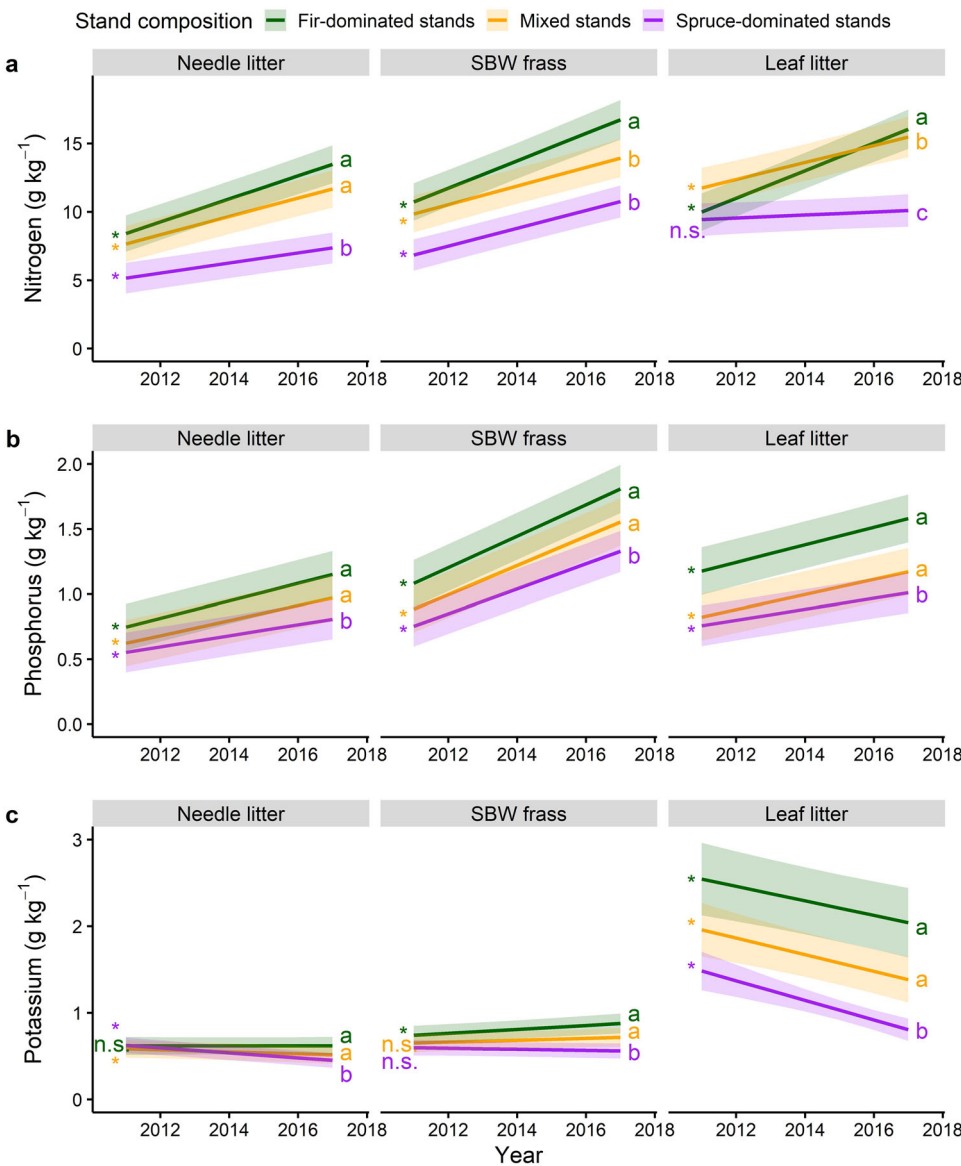

**Fig. 2 Nutrients concentration in litter with annual progression of the outbreak.** Fitted relationship for nitrogen (**a**), phosphorus (**b**), and potassium (**c**) concentrations (g kg⁻¹) with time (calendar years) and stand composition for different litter types. Shaded areas indicate 95% confidence intervals. Slopes significantly different from zero are represented by an asterisk and letters compare slopes among stand composition, separately for each litter type. Linear mixed-effects models were fitted using a total sample size of 1360 observations.

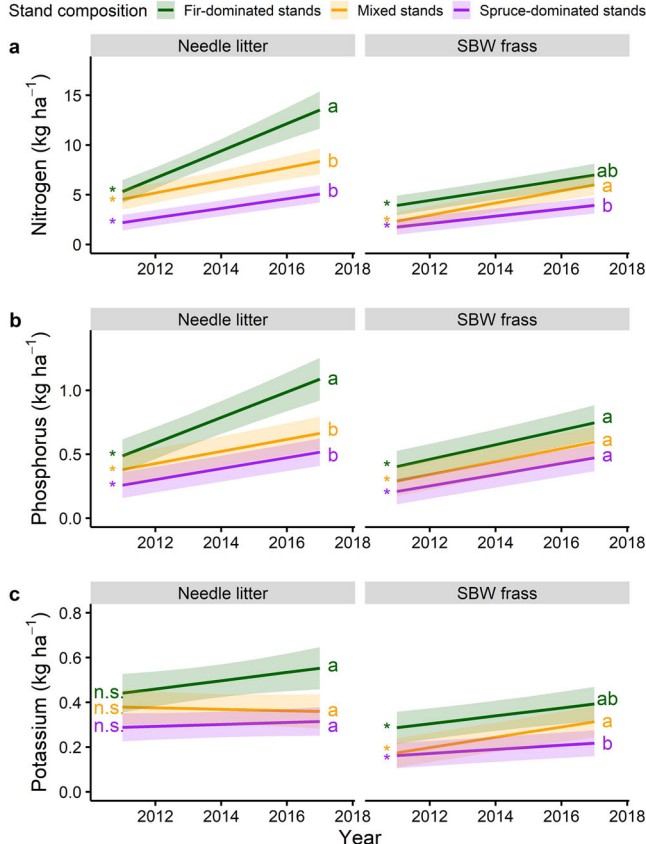

**Fig. 3 Litter nutrient fluxes with annual progression of the outbreak.**
Fitted relationship for annual nitrogen (**a**), phosphorus (**b**), and potassium (**c**) litter fluxes (kg ha$^{-1}$) with time (calendar years) and stand composition for different litter types. Shaded areas indicate 95% confidence intervals. Slopes significantly different from zero are represented by an asterisk and letters compare slopes among stand composition, separately for each litter type. Local basal area was held at its mean value (see Methods section). Linear mixed-effects models were fitted using a total sample size of 980 observations.

amounts reaches the forest floor in throughfall than in litter[35]. Insect damage to needles as well as increased throughfall, as the canopy thinned with outbreak progression, may have increased K leaching and decreased foliar litter concentrations. As a result, fluxes of K in needle litterfall did not vary significantly, as the outbreak progressed while enhanced fluxes in frass were mostly related to an increase in frass mass.

A significant increase in nutrient fluxes resulted from defoliation by the SBW, reaching in some cases more than five times the minimum level measured early in the outbreak. Considering that defoliation in the study area began as early as 2006 (5 years before the first measurements), it is conceivable that the increase in nutrient fluxes could be even greater. Defoliation and insect population build up significantly contributed to an increase in nutrient availability to forest soils as shown in other forest ecosystems[8,11,31]. Although there is strong annual variability in outbreak severity (Fig. 1), the build-up in nutrients resulted from not only the direct action of defoliation on litter fluxes, but also from its influence on forest conditions (e.g., canopy opening, tree mortality, and soil temperature) and nutrient dynamics. This idea is supported by the fact that the statistical model using calendar year performed better for most nutrients tested, compared to the one using annual larval abundance (see Methods). In the boreal forest biome, where substantial amounts of nutrients are locked up in organic structures in the soil, the effect of periodic pulses of defoliation can have important implications for recirculating nutrients and supporting long-term forest productivity[3,4].

Defoliation also improved litter quality in host needles, SBW frass litter types, as well as in non-host deciduous leaf litter. This nutrient-rich litter can rapidly transit to the forest floor and subsequently to above ground vegetation[6,10]. Insect cadavers and frass decompose more rapidly than forest litter derived from plants[36], particularly in boreal and subarctic forests where decomposition rates are low. Frass N quality is an important driver of soil N availability and N-rich plant leaves are strongly associated with N-rich frass[37,38] due to excess N in foliage being excreted by defoliating insects[39]. Furthermore, research has shown that herbivory enhanced litter quality by inducing abscission of damaged needles during the growing season[40]. It was estimated that for 1 g of foliage consumed by the SBW, 0.4 g was destroyed but not consumed and eventually reached the forest floor[41]. This improved litter nutrient quality and flux were accompanied by an increase in soil temperature with defoliation in all stand types, potentially speeding up decomposition and soil nutrient turnover[19–21]. This is consistent with a study that showed soil water content is enhanced by SBW defoliation because of lower canopy interception of rainfall and decreased evapotranspiration rates by the damaged canopy[42].

Improved nutrient fluxes from litterfall combined with favorable conditions for soil organic matter decomposition, i.e. increased soil temperature and possibly soil water content[43], enhanced soil nutrient supply for the surviving host trees and non-host vegetation. In addition, severe defoliation is known to trigger a strong increase in foliage production to compensate for loss of photosynthesis capacity in combination with longer foliage retention of older needles[44,45]. This suggests that nutrient demand in defoliated trees is maintained as defoliation progresses until tree starch reserves are too severely depleted and newly produced photosynthates can no longer support growth[46].

As expected, the increase in nutrient concentration was higher in photosynthetic tissues of the primary host of the SBW, balsam fir, and the quality and quantity of litter were greater in stands dominated by this species. Outbreak severity in these forests is closely linked to stand composition where stands dominated by balsam fir suffer more defoliation and mortality than stands dominated by black spruce[47]. However, although black spruce is a secondary host of the SBW, neighborhood effects with the primary host have been observed to increase the vulnerability of black spruce to the SBW[48]. We also observed that nutrient dynamics in stands composed of a mixture of both host species were closer to those observed in stands dominated by the primary host. Overall, plant communities in forest ecosystems dominated by balsam fir responded more strongly to this increase in nutrient availability, as revealed by enhanced nutrient contents in deciduous shrubs and tree leaves. In addition, we also observed a decrease in needle C:N ratio as the outbreak progressed, suggesting that the palatability of needles is enhanced by defoliation, as N concentration increases[49]. All these results point toward faster nutrient turnover in stands dominated by fir and greater defoliation by the insect in these forest types. This is also supported by the local increase in soil inorganic nitrogen availability when defoliation is high and fir dominates.

This study provides evidence that over the course of an outbreak, foliage nutrient quality increased with time in all stand types and tree species with potential positive feedbacks on defoliator feeding and development[50,51]. Plant diet quality, including differences among host species, has been identified as determining mortality rates of defoliators of hardwood trees in Europe by affecting feeding duration and larval growth[52]. Further, delayed induced resistance was observed in defoliated birch leaves that

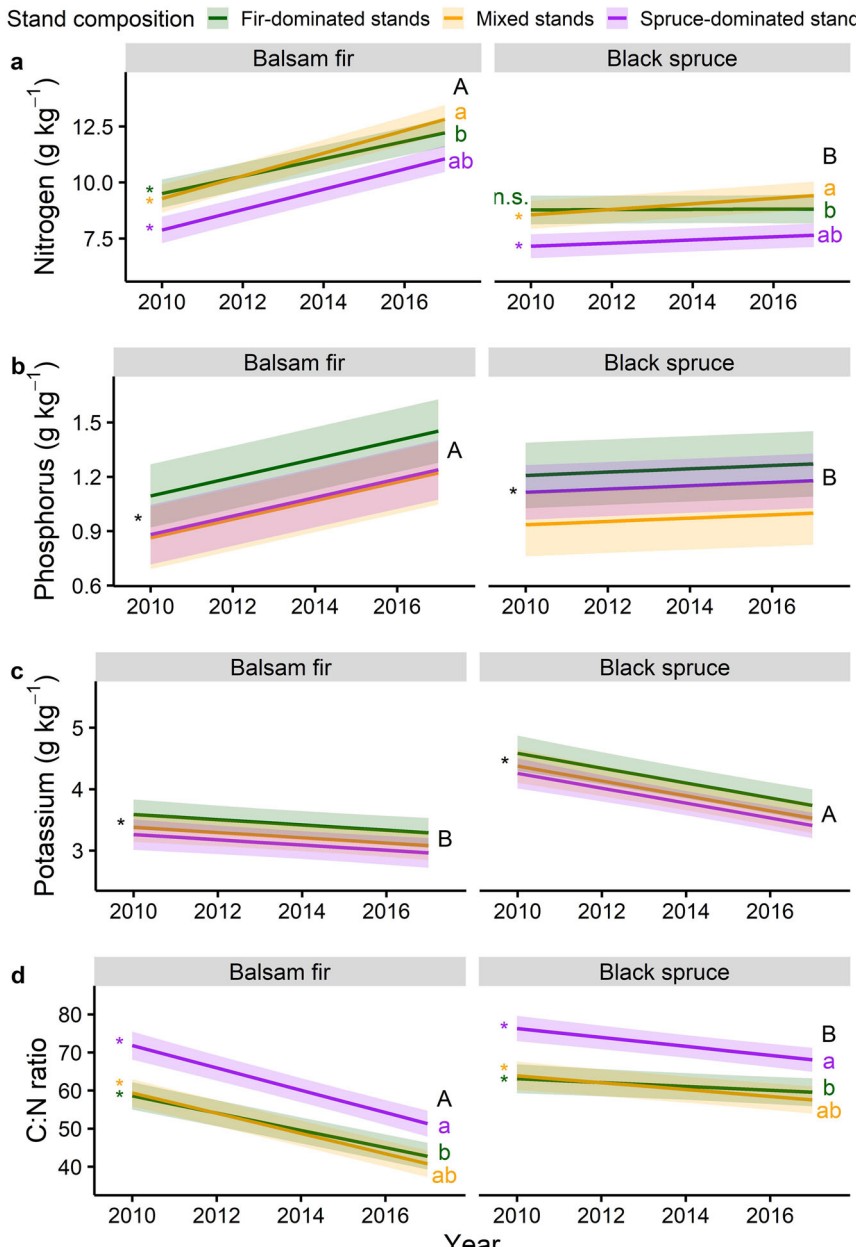

**Fig. 4 Nutrients concentration in foliar tissue of living SBW host trees with annual progression of the outbreak.** Fitted relationships for nitrogen (**a**), phosphorus (**b**), and potassium (**c**) concentrations (g kg$^{-1}$) and C:N ratio (**d**) with years (calendar years) and stand composition in needles taken off branches of living balsam fir and black spruce trees. Shaded areas indicate 95% confidence intervals. Slopes significantly different from zero are identified with an asterisk. Lowercase letters in color compare slopes among stand composition (when the interaction between year and stand type was included in the model) and uppercase letters in black compare slopes between species. Total sample size was 1693 observations.

reduced potential fecundity of defoliators[53]. In SBW, although balsam fir possessed high amounts of secondary compounds, larvae were able to consume more foliage from this host species compared to other hosts such as white spruce[54]. The SBW appears thus to have evolved compensatory mechanisms to counter defensive responses of its preferred host, balsam fir. Feeding duration is considered to be longer and defoliation is greater in balsam fir than in black spruce, because budburst of balsam fir occurs earlier and is better synchronized with SBW larval emergence in the spring[55]. In the present study, the effect of host species was observed for all categories of nutrient influx, as litter, frass and soil had higher nutrient fluxes in stands containing a higher proportion of balsam fir. Among multiple factors, the beginning of SBW outbreaks has been associated with masting

of balsam fir cones that provide a nutrient boost to young larvae[56] and increased larval survival in low-density populations. The transition from low population densities to broad-scale outbreaks has been a topic of debate for decades with no broad consensus among ecologists[26,57]. Mortality of young lepidopteran larvae is usually high and extreme variation in mortality can be an important factor in modulating SBW population dynamics[58,59]. A major mortality factor of SBW emerging from diapause in the spring is limited N availability[60]. If outbreaks begin in senescent stands nourished by breakdown of needle proteins[28,29] or from N-rich flowering cones[30], then feedbacks into the soil through nutrient fluxes caused by defoliation and their subsequent uptake by surviving defoliated trees can enhance the survival of young larvae. Increased survival of young larvae would sustain local

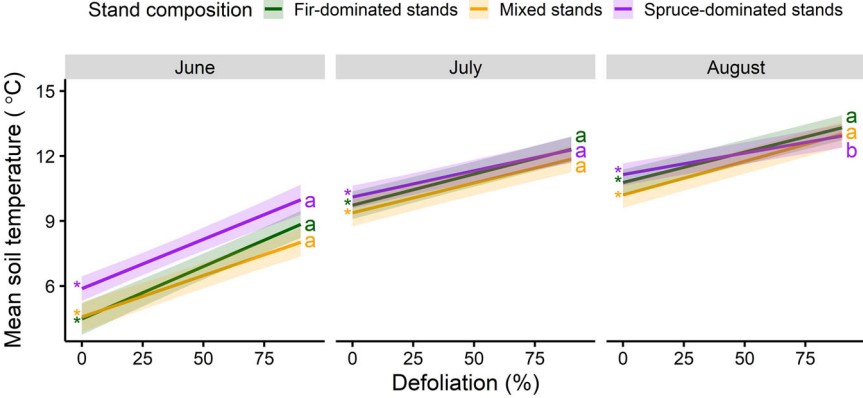

**Fig. 5 Change in soil temperature with annual progression of the outbreak.** Fitted relationships between soil temperature under the humus layer and local mean tree defoliation within a 5 m-radius and stand composition. Shaded areas indicate 95% confidence intervals. Slopes significantly different from zero are represented by an asterisk and letters compare slopes among stand composition types, separately for each month. The linear mixed-effect model was fitted on 3459 observations.

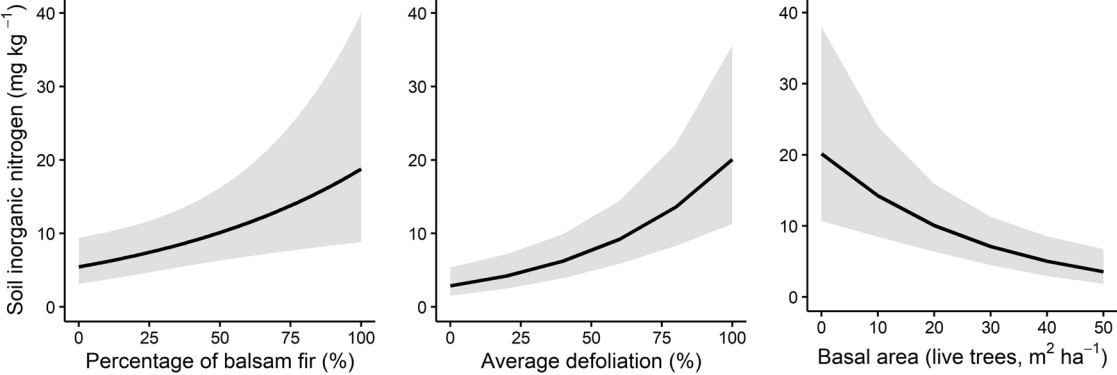

**Fig. 6 Soil inorganic Nitrogen concentration with annual progression of the outbreak.** Fitted relationships for soil inorganic nitrogen (SIN) concentration and proportion of balsam fir, average tree defoliation and basal area, within a 5 m-radius. Shaded areas indicate 95% confidence intervals. The generalized linear mixed model was fitted on 1548 observations.

populations long enough to increase insect fitness and population growth[29] to create "hot-spots" from which dispersing moths can drive the regional spread of outbreaks[61,62].

In the SBW system, we therefore distinguish two phases in the dynamics of nutrients. The first one is associated with the beginning of defoliation, when nutrients that fall to the ground can be largely captured by trees and increase the quality of their foliage. This phase could be closely related to the local increase in insect populations. After a certain threshold of cumulative defoliation is reached, the process of tree mortality begins, and the canopy is no longer able to capture nutrients, resulting in the observed increase in inorganic nitrogen in the soil[22]. At this point, massive adult dispersal will occur, and the outbreak will expand regionally to reach intact-forested landscapes[62]. This would cause lateral transfer of nutrients, estimated to be equivalent to or greater than the annual atmospheric deposition[63]. This regional pulse of nutrients could contribute to the insect's establishment success in these new forest landscapes.

In the boreal forest biome, where nutrients although abundant in the soil are largely immobilized and thus mostly unavailable to plants, defoliation by the SBW causes an increase in soil nutrient availability, which enhances the nutrient content of the foliage of surviving plants. Increased defoliation in the SBW-host tree system thus improves food quality, which acts as a form of facilitation in which net insect population growth rate increases with population density[28,29,60,64]. Nutrients made available from defoliator activity and changes in soil abiotic conditions

(e.g. temperature and water content) can be acquired by the surviving host and non-host plants, in turn benefiting defoliators by increasing the nutritional quality of host trees. Such feedback related to food quality and nutrient release has a strong potential for positive density dependence in nutrient-limited plant-herbivore systems[60,64] and is capable of supporting the growth of broad-scale outbreaks of defoliating insects such as the SBW. The greater the proportion of host plants in the ecosystem, the greater the likelihood that such a feedback loop would occur. This supports the role of bottom-up forces in contributing to the rising phase of outbreaks by allowing the insect to overcome population regulation from top-down pressure.

## Methods

**Experimental design.** The study area is in the North Shore region of the St. Lawrence River in the Province of Quebec, 50 km north of Baie-Comeau in one of the earliest epicenters of the current 13.5 million-ha outbreak. The climate is cold and maritime with a mean annual temperature of 1.7 °C and mean annual precipitation of 1040 mm[65]. Balsam fir and black spruce dominate the regional forest composition in pure and mixed stands[66,67]. SBW outbreaks occur periodically every 30–40 years over eastern Canada and the study region has undergone an increase in the severity and extent of outbreaks since the mid 20[th] century[68]. Balsam fir, the main host of the SBW, suffers severe defoliation and mortality as outbreak progresses, while black spruce is less severely defoliated, and suffers less mortality[47].

Within the study area, we established 10 permanent plots (size varying from 40 × 100 m to 60 × 100 m) at the onset of the outbreak (2006–2010): four in black spruce-dominated stands, three in balsam fir-dominated stands, and three in mixed balsam fir and black spruce-dominated stands[48]. All soils were podzolic. Half were

classified as Ferro-Humic Podzol and the other half as Humo-Ferric Podzol (CSSC 1998) with a thick (10 to 20 cm) FH humus layer. Soils were acidic with a surface mineral soil $pH_{(water)}$ averaging 4.5. Soil texture ranged from sandy loam to loamy sand.

Along a 70-m transect parallel to the longest axis (100 m) of each plot, we established seven sampling points (10 m apart) to measure annual litterfall (from 2011 to 2017), soil nutrient availability using ion-exchange resins (2010 or 2012 to 2017) and soil temperature (2011–2017). At each sampling location, three litter traps (40 cm × 60 cm) were laid ~1.5 m from a post at angles of 0°, 120°, and 240°, for 21 traps per plot. Traps were maintained all year and their content collected every year in early September, ~4–6 weeks after the end of defoliation. Soil nutrient availability was assessed with 20 g of mixed-bed ion exchange resins (Ionac NM-60, Lenntech, Delft, The Netherlands; $H^+/OH^-$ Form, Type I, Bead) that were placed in nylon bags, charged with 1 M HCl, and rinsed with deionized water. They were kept moist before being installed in the field. Two resin bags were installed in proximity to each litter trap in early August of year "$t$" and collected and replaced at the same period of year "$t+1$", for a total of 42 resin bags per plot. A soil temperature probe was inserted at the interface of the humus layer and the mineral soil in the proximity of each litter trap, for 21 probes per site. Within a 5-m radius around these seven sampling locations, basal area of SBW host trees was assessed in 2014, based on living stems with diameter at breast height (1.3 m from the ground) >5 cm. From 2012 to 2017, cumulative defoliation was visually estimated (in classes of 1–25%, 26–50%, 51–75%, and >75%) for each host tree present within the 5-m radius and averaged by sampling location.

**SBW population density and defoliation.** Two parallel transects were established on each side of the longest axis (100-m) of every plot. At 10-m intervals, a tree of both host species, when present, was sampled annually to evaluate the SBW population density and annual defoliation. In mid-June, when SBW individuals are at the fourth stage of larval development (L4), a 45 cm branch was cut at mid-crown with a pole pruner and brought back to the laboratory to count SBW larvae. Branches were weighed to standardize larval density and allow for comparisons among trees and stands. In mid-August, after the end of the defoliation period, a similar branch was sampled to measure annual percent defoliation using a proven methodology[69]. The SBW is a defoliator of annual foliage and thus it takes 5–7 years of severe defoliation before all foliage is removed. The same trees ($n = 360$) were followed for the entire study period or until their death. The remaining foliage from this sampling was kept for nutrient analyses.

**Sample processing.** Litter trap samples and host-tree needles were dried at 60 ºC for 72 hrs prior to sorting. Litter was sorted manually into five categories: leaves (shrubs and broad leaves), conifer needles, moss and lichen, branches and other material (mostly SBW frass). In this study, we only considered litter from deciduous leaves, needles and frass. Samples were weighed and then ground to 0.5 mm and a 2-g subsample was kept prior to chemical analysis. Concentration of total N and C were determined using a TruMac CNS Elemental Analyzer (LECO Corporation, St-Joseph, MI, USA). A subsample was reduced to ashes at 500 °C and then recovered in $1 \, mol \, L^{-1}$ HCl[70]. Phosphorus and K concentrations were determined by inductively coupled plasma analysis (PerkinElmer Optima 7300DV). Nutrient fluxes were obtained by multiplying the weight of each litter type sample by the concentration of the nutrient in the same sample. Host tree needles collected from branch sampling were processed the same way.

Resin $NH_4^+$ and $NO_3^-$ were extracted following the method described in Trottier-Picard et al.[71]. Briefly, resin bags were kept cool prior to analyses. In the lab, they were rinsed with deionized water, separated into two samples and weighed. Samples were extracted for 30 min in 50 mL of 2 M KCl. Concentrations of $NO_3^-$ and $NH_4^+$ in the KCl extracts were determined by flow injection (QuickChem 8500, Lachat Instruments, Loveland, CO). $NO_3^-$ and $NH_4^+$ were added together to become soil inorganic nitrogen (SIN). Due to machine calibration, small negative values are possible in the output; we replaced them by 0.001 (including observations equal to zero), a very small value compared to observed positive values that represent undetected SIN.

**Statistical analyses.** Linear least-squares regressions were used to verify whether average annual defoliation, larval density, annual litterfall, and cumulative tree mortality differed among stand composition types during the outbreak period. For the first three models, data for each stand were averaged over all sampled years. Tree mortality was not averaged as it is cumulative and thus the last year (2019) was selected for this model. Pairwise comparisons among all stand types were performed post hoc.

We used linear mixed-effects models to explore the relationships between nutrients in litterfall and outbreak progression (time) and to test whether this relationship differed with stand composition and among litter types. Models were fitted separately for each nutrient (N, P, and K) with either concentration or quantity of nutrients as the response variable. Litter type (needles, leaves and frass), stand composition (fir-dominated, mixed or spruce-dominated), year, their three-way interaction and all possible two-way interactions were considered as fixed effects. The quantity of leaf litter in each trap was insufficient for chemical analysis and were pooled at the sampling point level (three traps per sampling point),

resulting in seven samples per plot instead of 21. Therefore, nutrient concentrations and fluxes from all litter types were averaged at this level for consistency and comparability. Sampling points nested in sites were included as random intercepts to account for variation in nutrient concentration or quantity among sampling points and sites and repeated measurements at each sampling point. Annual nutrient flux (kg ha$^{-1}$) per litter type was estimated from the weight of the different litter types and nutrient concentrations, in proportion to the area covered by the traps (0.24 m$^2$). Broadleaf litter was excluded from the flux models due to its very low quantity in the traps. For these models, we also included the local basal area (see sampling design section above) to control for the effect of neighboring tree basal area on litterfall quantity.

A similar modeling structure was used to assess the effect of outbreak progression on tree foliage nutrients. Response variables were N, P, and K concentrations and C:N ratio in live needles. Stand composition, tree species, year, and their interactions were evaluated as possible fixed effects while sampled trees nested within a site were included as random intercepts to account for variation in needle nutrient concentration among sites and trees as well as repeated measurements on the same trees.

A linear mixed-effect model was also used to test for changes in summer soil temperature with increasing tree defoliation. Average annual percent defoliation (median of defoliation classes) in a 5-meter radius around each sample point in interaction with month (June, July, and August) and stand composition were used as fixed effects and temperature probes nested in sampling points and site as random effects.

As soil inorganic N concentration (SIN) shows very high spatial variability, variables related to outbreak progression (forest composition and tree defoliation) were measured at a finer scale in this analysis. Mean cumulative defoliation, fir abundance and live basal area were calculated in a 5 m radius around each sampling point (where the resin bags were installed). A generalized linear mixed model (GLMM) with a gamma distribution and log link function was fitted with SIN as the response variable, defoliation, percentage of fir, and basal area as fixed effects and sampling points nested in sites as random effects.

To simplify presentation of the results, interactions that were not statistically significant, when tested, were removed from the models (however when a three-way interaction was significant, all two-way interactions were kept in the models regardless of their significance). In preliminary analyses, we assessed whether using SBW population data (mean number of 4th instar larvae, L4s, per kg of branch in each site), instead of year to express outbreak progression in the nutrient models, provided a better fit. Although it did for some response variables, the majority had the lowest AIC when using year, therefore it was chosen for all models.

All statistical analyses were performed using the R software version 4.1.2 [72]. Models were fitted using the *lm* function for least-squares regressions, *lme* function from the nlme package[73] for linear mixed-effects models and the *glmer* function from the lme4 package[74] for the generalized mixed-effects model. Model assumptions were checked using plots of residuals. In the linear mixed models, a power variance function structure with fitted values as a variance covariate was used to achieve homoscedasticity, except for the soil temperature model where a constant variance function structure with month as a grouping factor provided the best fit. Tests on slopes, calculations of estimated marginal means (EMMs) and pairwise comparisons among stand types were performed post hoc using the emmeans package[75] and adjusted for simultaneous inference using the "mvt" method.

**Reporting summary.** Further information on research design is available in the Nature Research Reporting Summary linked to this article.

## Data availability
All data are available on Canada's Open Government Portal https://doi.org/10.23687/278a6e45-508f-485c-bb2f-41cc305f302a Data formatted to produce Figs. 1–6 can be found in Supplementary Data 1.

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

## Acknowledgements

The authors thank two anonymous reviewers and the associate editor for insightful comments, which improved the quality of the manuscript. The authors are grateful to M. Gauvin, H. Dorion, C. Moulin-Vézina, C. Simard, and many field assistants for their help in collecting and processing the samples. We thank S. Rousseau for the chemical analyses of all the samples. Natural Resources Canada, Canadian Forest Service provided funding for this work (2010–2018).

## Author contributions

L.D.G., D.P., M.S., J.M.G., D.D.K., and D.S.P. designed the study and methodology. D.G. and S.B. performed the sampling and processed the data. M.M. and D.B. performed the statistical analyses. L.D.G., D.S.P., D.P., D.D.K. wrote the original draft and all authors reviewed, commented, and edited draft versions.

## Competing interests

The authors declare no competing interests.
