## [Peer Review File · Communications Biology]

Reviewers' comments:

Reviewer #1 (Remarks to the Author):

The manuscript by De Grandpre et al. investigates the effects of defoliation on nutrient cycling in a forest ecosystem. The authors find that during the process of defoliation by the spruce budworm (SBW) nutrient availability in the soil (e.g., needle and leaf litter) increases, which results in an increase in nutrients in leaves. As a result, the authors argue, causes an increase in SBW populations leading to outbreaks. In my view, this manuscript provides novel evidence of positive bottom-up feedbacks from defoliation that may explain insect outbreaks.

Overall, the manuscript is well written. The data collected is intensive and over a period of 7 years. The data analyses seem appropriate. Although the conclusions are in general supported by the data, I have a few comments that (I hope) may help improve the quality of the manuscript.

General comments:

1. Figure 1A-C shows a decrease in defoliation, larval density, and litterfall from 2014-2016 even when nutrient availability in the soil and leaf quality increased. Why? Any explanation from this drop?
2. Also, potassium tended to decrease in the soil (leaf litter) and leaves through the process of defoliation. Any explanation why?
3. The authors do not report any statistical analysis for the data on annual defoliation, leaf density, litterfall, and tree mortality. I wonder why? These data need to be analyzed statistically to show if there are significant differences among stand composition.
4. Throughout the manuscript the authors use both common names and scientific names of the trees. Please be consistent.
5. It was my belief that outbreak cycles are influenced by the trees' induced defenses such as, as the outbreak progresses, not only nutrients might be increasing but also the allocation to plant defenses. I thought that increases in induced defenses ("delayed induced changes") causes the pest populations to eventually collapse. For example, see "Delayed Induced Changes in the Biochemical Composition of Host Plant Leaves during an Insect Outbreak" by Pekka Kaitaniemi, Kai Ruohomäki, Vladimir Ossipov, Erkki Haukioja and Kalevi Pihlaja. *Oecologia*. Vol. 116, No. 1/2 (1998), pp. 182-190. This needs to be discussed in the context of the authors' findings. The authors only measured nutrient availability but not secondary metabolites (induced defenses), which may also play a role in regulating the pest populations.
6. It is always helpful to include line numbers to help with reviewers' comments.

Specific comments:

Page 2, line 3: "insect-caused"

Page 4, line 1: "bottom-up forces"

Page 4, line 20: add common name since it is used throughout the paper: "balsam fir, *Abies balsamea*"

Page 4, line 21: add common name "black spruce, *Picea mariana*"

Page 5, line 6: delete "and" after "(3)". Already mentioned.

Page 5, line 6: change to "inputs, the rise"

Page 5, line 7: "changes in soil" not "changes in soil"

Page 5, lines 12-21: how come these data were not analyzed statistically?

Page 6, lines 1-6: how come these data were not analyzed statistically?

Page 6, lines 19: any ecological consequences for the observed decrease in K? This is not discussed.

Page 7, line 5: do not italicize "foliage"

Page 7, line 6: I think it reads better as "in balsam fir-dominated stands"

Page 8, line 13: "seven years of our study (2010-2017)"

Page 10, line 18: "*A. balsamea*"

Page 11, line 8: for consistency, "SBW" instead of "spruce budworm"

Page 11, line 13: "SBW" instead of "spruce budworm"

Page 11, lines 13-14: "SBW" instead of "spruce budworm"

Page 12, line 19: "SBW" instead of "spruce budworm"

Page 12, line 22: "top-down pressure"? Not sure what the authors mean by "top-down limitations"

Page 13, line 10: italicize "Picea mariana"

References

Please check all references to comply with journal guidelines.

Swank et al., Lovett et al., Forkner and Hunter, Fierravanti et al., Martineau et al., McMillin and Wagner, Mattson - do not use capital letters in all title words

Le Mellec and Michalzik – italicize scientific name

Reviewer #2 (Remarks to the Author):

COMMUNICATIONS BIOLOGY

Title: Defoliation-induced changes in foliage quality trigger broad-scale insect outbreaks

This study presents a very interesting body of work, investigating the relationship between bioavailability and flux of nutrients and the progression of a herbivore outbreak on primary and secondary hosts in a boreal forest. The authors present data suggesting that increasing defoliation through the herbivore creates a positive feedback loop that releases even higher rates of nutrients into the ecosystem of which a considerable part is funneled back into the infested trees. They state that, in an ecosystem like boreal forests where a major fraction of nutrients is inaccessible in complex organic compounds, such altered nutrient dynamics can accelerate the herbivore outbreak dynamics.

This approach is very interesting and the authors present data that supports this hypothesis. In my opinion they should, however, be a bit more careful in their wording when claiming to "provide evidence" (e.g. in the abstract). This implies a causality that I am not quite comfortable with.

1. The manuscript is well written but it would be much easier to follow if the authors chose to use terms more consistently. For instance, they are using four different terms each for the two tree species studied: the scientific name, an abbreviation of the scientific term, the common name and an abbreviation of the common name. I suggest choosing one and sticking to it, both in the text and in the figures. (I personally like the version in the figures 1-5: firs and spruces in fir- and spruce-dominated stands.)

Introduction:

2. In hypothesis 3 you mention soil temperature the first time, without presenting information on its relevance beforehand. Please add the necessary information in the introduction.

Results:

3. In many cases you state that one treatment was higher without mentioning the compared counterpart (higher than what?). Please check this and add the information.

4. The results section would also be easier to understand if the same trends in the data were described with the same sentence structure. In fig. 3, for example, you always have the highest values in the fir-dominated stands but in the description you change between stating that the fir-dominated stands had higher concentrations and that the spruce-dominated stands had lower concentrations. It would be easier if you would stick to stating which one is higher.

5. Also, you are jumping between figures 2 and 3 in the description. It would be easier to follow if you finished one figure and then continued to the next. (If you want to describe one nutrient after the other I suggest you compile the figures accordingly.)

6. Page 7 and fig. 5: you state that June warming rates were similar: does that mean that there was no significant difference? If yes, please write that. If there are no statistically significant differences, strictly speaking you cannot add letters to the graphs.

7. Maybe you could transfer the fig legend on top of each figure as it refers to all the presented

graphs and not just the one where it is inserted.

8. In fig. 4, the letters indicating significant differences are referring to both species, correct? This is a little confusing as it was different in the previous figures.

9. In fig. 5, please specify that it is the soil temperature that is presented

10. Could you please elucidate why the results on SIN could not be evaluated statistically? You obviously do have values for SIN depending on degree of defoliation and fraction of balsam fir in the stands.

Discussion:

11. Page. 9: please elucidate how needle abscission during the growing season improves litter quality. Is this statement related to your results or to the cited reference (36)? Where did you show that a higher than usual fraction of needles was falling from the herbivore-infested trees during growing season?

12. Please add a reference for the sped up decomposition and nutrient turnover in warmer soils. How are the results on soil temperature consistent with results on soil water?

13. Please explain what happened in 2015 that lead to such a severe drop in defoliation and larval density as shown in fig. 1.

14. In fig. 2, K concentration in frass is not following the development of K concentration in needles while that is indeed the case concerning P and N concentration. Please elucidate.

Methods:

15. Page 16: I imagine "the quantity of nutrients dropped on the floor" is the nutrient flux? Please state this explicitly and describe in detail how you calculated the nutrient flux.

Supplemental tables:

16. I find it difficult to follow which results are significantly different from what. (Especially regarding the pairwise comparison in tab. S5.) For readers who are not familiar with the specific statistical method, it might be a bit confusing.

17. Maybe you could change the table structure to more clearly present the relevant values and indicate their respective significant differences in a way that is easily comprehensible? (What about using letters instead of asterisks?)

We have reviewed the manuscript according to all the comments made by the two reviewers. In the revised manuscript, the modifications we made appear in red.

Reviewer #1 (Remarks to the Author):

The manuscript by De Grandpre et al. investigates the effects of defoliation on nutrient cycling in a forest ecosystem. The authors find that during the process of defoliation by the spruce budworm (SBW) nutrient availability in the soil (e.g., needle and leaf litter) increases, which results in an increase in nutrients in leaves. As a result, the authors argue, causes an increase in SBW populations leading to outbreaks. In my view, this manuscript provides novel evidence of positive bottom-up feedbacks from defoliation that may explain insect outbreaks.

Overall, the manuscript is well written. The data collected is intensive and over a period of 7 years. The data analyses seem appropriate. Although the conclusions are in general supported by the data, I have a few comments that (I hope) may help improve the quality of the manuscript.

Thanks for the review and the constructive comments. We have addressed all the comments in our review and we do agree that it contributed to improve the quality of the manuscript.

General comments:

1. Figure 1A-C shows a decrease in defoliation, larval density, and litterfall from 2014-2016 even when nutrient availability in the soil and leaf quality increased. Why? Any explanation from this drop?

The decrease in defoliation, larval density and litterfall in 2015 and 2016 followed a peak of these variables in 2014. As SBW larvae feed almost entirely on current year needles, such fluctuations in SBW population are often observed locally. After several years of severe defoliation, it is common to observe local population collapse of the insect, as weakened hosts do not produce enough current year foliage to maintain viable SBW populations. At this time, SBW will disperse locally to find suitable hosts, and in years of high regional populations density and severe defoliation, it can disperse over very long distances (>100km-500km) as they are transported by wind currents at altitudes of few hundred meters. In 2014, defoliation was very severe over a vast area (>5 000 km²) and SBW populations density were very high after several years of population build-up. That year, the region under study was a source of moths, and long-distance dispersal was observed. The following 2 years SBW populations were low in the region and host trees that were still alive were able to invest in growth and put a lot of energy in photosynthetic material. The region then became a sink of moths explaining the rise in larval density and the increase in defoliation in 2017. In the case of litterfall, although we observed an annual pattern similar to that of defoliation, the drop in 2015 and 2016 was not as sharp. As the pressure of defoliation decreased, trees invested in growth and new photosynthetic material. During that period, shedding of older foliage may have increased. This is commonly observed following stresses such as drought stress.

As for the linear increase in nutrient flux and quality, this relates to the linear modelling approach we used. To consider the annual fluctuations in the outbreak on nutrient dynamics, we tested a mixed model with the number of SBW larvae per kg of branches and compared it to the one where calendar year was used. The model using calendar year was performing better than the one with number of larvae. This suggests that even if there is strong annual variability in outbreak severity, the build-up in nutrients is probably multifactorial in causes and its importance changes with time as the forest conditions are modified by the action of the insect.

We addressed this in the manuscript (lines 197-202).

Although there is strong annual variability in outbreak severity (Fig. 1), the build-up in nutrients resulted from not only the direct action of defoliation on litter fluxes, but also from its influence on forest conditions (e.g., canopy opening, tree mortality, and soil temperature) and nutrient dynamics. This idea is supported by the fact that the statistical model using calendar year performed better for most nutrients tested, compared to the one using annual larval abundance (see Methods).

2. Also, potassium tended to decrease in the soil (leaf litter) and leaves through the process of defoliation. Any explanation why?

Potassium is an element that is cycled wastefully at the plant level (Schlesinger 2021). Much more potassium cycles in throughfall than in litterfall, up to 65 times more according to some studies (Schlesinger 2020). For this element, litterfall represents only a small portion of the plant to soil flux.

We added to the discussion to explain this pattern (lines 183-190)

The only exception is K for which the concentration in live needles and deciduous leaf litter decreased over the course of the outbreak. Potassium is an element that is cycled wastefully at the plant level, it is easily leached from foliage and much greater amounts reaches the forest floor in throughfall than in litter³⁵. Insect damage to needles as well as increased throughfall, as the canopy thinned with outbreak progression, may have increased K leaching and decreased foliar litter concentrations. As a result, fluxes of K in needle litterfall did not vary significantly, as the outbreak progressed while enhanced fluxes in frass were mostly related to an increase in frass mass.

Schlesinger, W.H. Some thoughts on the biogeochemical cycling of potassium in terrestrial ecosystems. *Biogeochemistry* **154**, 427–432 (2021). <https://doi.org/10.1007/s10533-020-00704-4>

3. The authors do not report any statistical analysis for the data on annual defoliation, leaf density, litterfall, and tree mortality. I wonder why? These data need to be analyzed statistically to show if there are significant differences among stand composition.

Although this was not the objective of the study, we agree with the reviewer. We now include some stats to explain the main differences. Analyses were performed to stress out differences among stand composition. These new results are presented in the beginning of the results section (lines 117-129). We also provided the statistics in Supplementary Table 1.

4. Throughout the manuscript the authors use both common names and scientific names of the trees. Please be consistent.

We agree that this was confusing, and we have now corrected the issue in all the manuscript. We now use common name for species and stands associated to the dominant species.

5. It was my belief that outbreak cycles are influenced by the trees' induced defenses such as, as the outbreak progresses, not only nutrients might be increasing but also the allocation to plant defenses. I thought that increases in induced defenses ("delayed induced changes") causes the pest populations to

eventually collapse. For example, see “Delayed Induced Changes in the Biochemical Composition of Host Plant Leaves during an Insect Outbreak” by Pekka Kaitaniemi, Kai Ruohomäki, Vladimir Ossipov, Erkki Haukioja and Kalevi Pihlaja. *Oecologia*. Vol. 116, No. 1/2 (1998), pp. 182-190. This needs to be discussed in the context of the authors’ findings. The authors only measured nutrient availability but not secondary metabolites (induced defenses), which may also play a role in regulating the pest populations.

Balsam fir produces high amounts of secondary metabolites during herbivory. This does not prevent SBW from feeding on it. From Fuentealba and Bauce (2016)

<http://dx.doi.org/10.1016/j.actao.2015.11.001> *“Balsam fir exhibited higher foliar toxic secondary compounds concentrations than white spruce in all drainage classes, resulting in lower male pupal mass, survival and longer male developmental time. This, however, did not prevent spruce budworm from consuming more foliage in balsam fir than in white spruce. This response suggests that either natural levels of measured secondary compounds do not provide sufficient toxicity to reduce defoliation, or spruce budworm has developed compensatory mechanisms, which allow it to utilize food resources more efficiently or minimize the toxic effects that are produced by its host’s defensive compounds.”*

Furthermore, most of secondary metabolites in both balsam fir and black spruce are monoterpenes, which are C based molecules. On the other hand, SBW populations collapse were shown to result from a combination of reduction in host foliage and a top-down control of parasitoids (Pureswaran et al. 2016).

We added something in the discussion (lines 250-255)

Further, delayed induced resistance was observed in defoliated birch leaves that reduced potential fecundity of defoliators⁵³. In SBW, although balsam fir possessed high amounts of secondary compounds, larvae were able to consume more foliage from this host species compared to other hosts such as white spruce⁵⁴. The SBW appears thus to have evolved compensatory mechanisms to counter defensive responses of its preferred host, balsam fir.

6. It is always helpful to include line numbers to help with reviewers’ comments.

You are absolutely right.

Specific comments:

Page 2, line 3: “insect-caused” - Done

Page 4, line 1: “bottom-up forces” - Done

Page 4, line 20: add common name since it is used throughout the paper: “balsam fir, *Abies balsamea*” - Done

Page 4, line 21: add common name “black spruce, *Picea mariana*” - Done

Page 5, line 6: delete “and” after “(3)”. Already mentioned. - Done

Page 5, line 6: change to “inputs, the rise” - Done

Page 5, line 7: “changes in soil” not “changes is soil” - Done

Page 5, lines 12-21: how come these data were not analyzed statistically? Added

Page 6, lines 1-6: how come these data were not analyzed statistically?

We just wanted first to present the general trends in nutrient fluxes (min-max). The GLM were done on each nutrient trend (concentration and fluxes) through time and presented in the following paragraphs.

Page 6, lines 19: any ecological consequences for the observed decrease in K? This is not discussed.

Done

Page 7, line 5: do not italicize "foliage" - Done

Page 7, line 6: I think it reads better as "in balsam fir-dominated stands" - Done

Page 8, line 13: "seven years of our study (2010-2017)" - Done

Page 10, line 18: "A. balsamea" - We decided to use common name

Page 11, line 8: for consistency, "SBW" instead of "spruce budworm" - Done

Page 11, line 13: "SBW" instead of "spruce budworm" - Done

Page 11, lines 13-14: "SBW" instead of "spruce budworm" - Done

Page 12, line 19: "SBW" instead of "spruce budworm" - Done

Page 12, line 22: "top-down pressure"? Not sure what the authors mean by "top-down limitations" -

Done

Page 13, line 10: italicize "Picea mariana" - Replaced by "black spruce"

References

Please check all references to comply with journal guidelines.

Swank et al., Lovett et al., Forkner and Hunter, Fierravanti et al., Martineau et al., McMillin and Wagner,

Mattson - do not use capital letters in all title words - Done

Le Mellec and Michalzik – italicize scientific name - Done

Reviewer #2 (Remarks to the Author):

COMMUNICATIONS BIOLOGY

Title: Defoliation-induced changes in foliage quality trigger broad-scale insect outbreaks

This study presents a very interesting body of work, investigating the relationship between bioavailability and flux of nutrients and the progression of a herbivore outbreak on primary and secondary hosts in a boreal forest. The authors present data suggesting that increasing defoliation through the herbivore creates a positive feedback loop that releases even higher rates of nutrients into the ecosystem of which a considerable part is funneled back into the infested trees. They state that, in an ecosystem like boreal forests where a major fraction of nutrients is inaccessible in complex organic compounds, such altered nutrient dynamics can accelerate the herbivore outbreak dynamics.

This approach is very interesting and the authors present data that supports this hypothesis. In my opinion they should, however, be a bit more careful in their wording when claiming to "provide evidence" (e.g. in the abstract). This implies a causality that I am not quite comfortable with.

We appreciate the positive comments of the reviewer. We agree with the reviewer that our choice of words was implying causality and we changed it to: **Our results suggest that a positive bottom-up feedback to forest ecosystems from defoliation may result in conditions favorable to self-amplifying population dynamics in insect herbivores that can contribute to driving broad-scale outbreaks. (Lines: 52-55). We also modified the title to take into account the comment of the reviewer.**

1. The manuscript is well written but it would be much easier to follow if the authors chose to use terms

more consistently. For instance, they are using four different terms each for the two tree species studied: the scientific name, an abbreviation of the scientific term, the common name and an abbreviation of the common name. I suggest choosing one and sticking to it, both in the text and in the figures. (I personally like the version in the figures 1-5: firs and spruces in fir- and spruce-dominated stands.)

Agree. We now use consistently the common name for the tree species and the abbreviation SBW for the spruce budworm.

Introduction:

2. In hypothesis 3 you mention soil temperature the first time, without presenting information on its relevance beforehand. Please add the necessary information in the introduction.

Some background information on soil temperature was added in the introduction (lines 73-78).

Increase in soil temperature following disturbance promotes microbial activity and results in a positive effect on nutrient cycling¹⁹⁻²¹. Such major shifts in nutrient fluxes are usually observed following broad-scale host-tree mortality during outbreaks^{18,22,23}. In defoliation episodes, it is likely that surviving host trees benefit from the greater supply of available soil nutrients as the canopy progressively opens and forest environmental conditions, such as soil temperature, are modified^{5,10,12,20,24}.

Results:

3. In many cases you state that one treatment was higher without mentioning the compared counterpart (higher than what?). Please check this and add the information.

This was modified throughout the results section.

4. The results section would also be easier to understand if the same trends in the data were described with the same sentence structure. In fig. 3, for example, you always have the highest values in the fir-dominated stands but in the description, you change between stating that the fir-dominated stands had higher concentrations and that the spruce-dominated stands had lower concentrations. It would be easier if you would stick to stating which one is higher.

This was modified throughout the results section.

5. Also, you are jumping between figures 2 and 3 in the description. It would be easier to follow if you finished one figure and then continued to the next. (If you want to describe one nutrient after the other I suggest you compile the figures accordingly.)

We changed the structure of the result section to describe one figure after the other.

6. Page 7 and fig. 5: you state that June warming rates were similar: does that mean that there was no

significant difference? If yes, please write that. If there are no statistically significant differences, strictly speaking you cannot add letters to the graphs.

The text was modified to reflect that there were no significant differences in the warming rate between stand types. We kept the letters because there were significant differences among stand types in August (Lines 167-170).

7. Maybe you could transfer the fig legend on top of each figure as it refers to all the presented graphs and not just the one where it is inserted.

This was done

8. In fig. 4, the letters indicating significant differences are referring to both species, correct? This is a little confusing as it was different in the previous figures.

This is correct. We have modified how we presented the comparisons among stand types and between species and added some information in the figure legend in order to make this clear.

9. In fig. 5, please specify that it is the soil temperature that is presented

Done

10. Could you please elucidate why the results on SIN could not be evaluated statistically? You obviously do have values for SIN depending on degree of defoliation and fraction of balsam fir in the stands.

We have addressed this comment. Although we did not want to put too much focus on these results, we do agree that it is better that they are presented more rigorously. With a GLMM approach using a Gamma distribution to accommodate non-normally distributed residuals, we were able to illustrate the significant trends in soil inorganic nitrogen as the outbreak progressed. However, we only considered data from 2014 to 2017, because in the beginning of the outbreak (2011-2013) there were not enough changes in the forest conditions (e.g. canopy opening) to generate a strong and consistent response in SIN. In addition, it is only from 2014 that we have data from all the 10 plots.

See results (lines 171-173), the new Figure 6 and methods (lines 398-404)

Discussion:

11. Page. 9: please elucidate how needle abscission during the growing season improves litter quality. Is this statement related to your results or to the cited reference (36)? Where did you show that a higher than usual fraction of needles was falling from the herbivore-infested trees during growing season?

This was clarified in the discussion, and we added a sentence and a reference to support the wasteful feeding habit of the SBW. See lines 212-215

Furthermore, research has shown that herbivory enhanced litter quality by inducing abscission of damaged needles during the growing season⁴⁰. It was estimated that for 1 g of foliage consumed by the SBW, 0.4 g was destroyed but not consumed and eventually reached the forest floor⁴¹.

12. Please add a reference for the sped up decomposition and nutrient turnover in warmer soils. How are the results on soil temperature consistent with results on soil water?

We added references to support this statement (line 217)

13. Please explain what happened in 2015 that lead to such a severe drop in defoliation and larval density as shown in fig. 1.

See lines 197-202 and response to comment 1 of the first reviewer.

14. In fig. 2, K concentration in frass is not following the development of K concentration in needles while that is indeed the case concerning P and N concentration. Please elucidate.

See response to comment 2 of the first reviewer. We also addressed it in the discussion (lines 183-190).

Methods:

15. Page 16: I imagine “the quantity of nutrients dropped on the floor” is the nutrient flux? Please state this explicitly and describe in detail how you calculated the nutrient flux.

We now use nutrient flux instead of the prior and we added some precisions on how it was calculated (see lines 345-355).

Supplemental tables:

16. I find it difficult to follow which results are significantly different from what. (Especially regarding the pairwise comparison in tab. S5.) For readers who are not familiar with the specific statistical method, it might be a bit confusing.

We have clarified in the legend and inside the table what the results presented in each table represent. Part of the description was missing in the Supplementary Table 5 (now labeled Supplementary Table 6), which we agree was confusing. We corrected this mistake.

17. Maybe you could change the table structure to more clearly present the relevant values and indicate their respective significant differences in a way that is easily comprehensible? (What about using letters instead of asterisks?)

The same results as those presented in the Supplementary tables can be seen differently in the figures, with letters indicating statistically significant differences, and relevant values are underlined in the Results section of the manuscript. These tables are presented in Supplementary information merely for readers interested in seeing exact coefficient values with their confidence intervals. We did not change the table structure but added clarifications in the table legends.

REVIEWERS' COMMENTS:

Reviewer #1 (Remarks to the Author):

I have read the responses of the authors to my comments, and I am satisfied with the way that they have addressed them in the new version of the manuscript.

Reviewer #2 (Remarks to the Author):

The authors have carefully considered and addressed my suggestions and concerns. I have no further comments and I'm happily recommending this interesting research for publication.

REVIEWERS' COMMENTS

Reviewer #1 (Remarks to the Author):

I have read the responses of the authors to my comments, and I am satisfied with the way that they have addressed them in the new version of the manuscript.

Reviewer #2 (Remarks to the Author):

The authors have carefully considered and addressed my suggestions and concerns. I have no further comments and I'm happily recommending this interesting research for publication.

We have addressed all suggestions to the satisfaction of the reviewers.